# Imeglimin Alleviates High-Glucose-Induced Bioenergetic and Oxidative Stress Thereby Enhancing Intercellular Adhesion in H9c2 Cardiomyoblasts

**DOI:** 10.3390/ijms26188913

**Published:** 2025-09-12

**Authors:** Hiroshi Ohguro, Megumi Watanabe, Megumi Suzuki, Naruki Ohara, Toshifumi Ogawa, Tatsuya Sato, Toshiyuki Yano

**Affiliations:** 1Departments of Ophthalmology, Sapporo Medical University, S1W17, Chuo-ku, Sapporo 060-8556, Japan; watanabe@sapmed.ac.jp (M.W.); sapmed15m018@gmail.com (N.O.); 2Departments of Cellular Physiology and Signal Transduction, Sapporo Medical University, S1W17, Chuo-ku, Sapporo 060-8556, Japan; a08m024@yahoo.co.jp; 3Departments of Cardiovascular, Renal and Metabolic Medicine, Sapporo Medical University, S1W17, Chuo-ku, Sapporo 060-8556, Japan; oltomwaits55@gmail.com

**Keywords:** H9c2 cell, metformin, imeglimin, extracellular flux analyzer, ROS

## Abstract

To elucidate the effects of the new antidiabetic agent, imeglimin (Ime, 2 mM), on high-glucose-induced cellular stress in cardiac cells, its effects were compared with those of the conventional antidiabetic agent metformin (Met, 2 mM) based on various cellular pathophysiological functions. H9c2 cardiomyoblasts were cultured under normal-glucose (5.5 mM, N-Glu) or high-glucose (50 mM, H-Glu) conditions. Cellular metabolic function was evaluated using a Seahorse XFe96 Bioanalyzer, along with measurements of reactive oxygen species (ROS) production, expression levels of the autophagy-related marker LC3, and intercellular adhesion properties measured based on transepithelial electrical resistance (TEER). Cells cultured under H-Glu conditions showed enhanced mitochondrial and glycolytic activities, which were suppressed by Met or Ime. Under H-Glu conditions, total cellular ROS (t-ROS) levels were significantly increased. Met had little effect on t-ROS under H-Glu conditions, whereas Ime markedly reduced both t-ROS and mitochondrial ROS (m-ROS) levels under H-Glu conditions. The LC3-II/LC3-I ratio, a marker of autophagic activity, decreased under H-Glu conditions; however, this reduction was not significantly affected by treatment with either Met or Ime. Regarding intercellular adhesion properties, TEER values were elevated under H-Glu conditions compared to N-Glu conditions, and those under H-Glu conditions were further increased by Ime but not Met. In support of these results, the mRNA levels of cell-adhesion-related molecules, including β-catenin and N-cadherin, were also altered by Ime. Collectively, Ime modulated high-glucose-induced alterations in the biological properties of H9c2 cardiomyoblasts, independent of changes in autophagic activity.

## 1. Introduction

Diabetes is characterized by elevated circulating glucose and fatty acid (FA) levels, leading to substantial metabolic remodeling in various organs, including the heart. Altered substrate availability in the myocardium plays a key role in impairing cardiac energy metabolism and promoting the excessive generation of reactive oxygen species (ROS), which can contribute to the development of diabetes-induced cardiac dysfunction [1,2,3,4]. Alterations in myocardial substrate utilization, including a shift toward FA oxidation, have been implicated in cardiac dysfunction via mechanisms such as the Randle cycle [3,4]; however, emerging evidence indicates that glucose overload under hyperglycemic conditions also promotes ROS generation, leading to oxidative stress [5], which results in structural and functional alterations in the myocardial tissue [6,7]. Therefore, augmented ROS levels and subsequent oxidative stress related to chronic hyperglycemia are major pathogenic factors and rational therapeutic targets for the development and progression of diabetes-induced cardiac dysfunction [8,9,10,11].

Metformin (Met) is the most commonly used glucose-lowering agent for patients with type 2 diabetes (T2D) primarily because of its ability to suppress hepatic gluconeogenesis via an adenosine 5′-monophosphate (AMP)-activated protein kinase (AMPK)-dependent mechanism [12,13]. Notably, in addition to its glucose-lowering action, Met exerts AMPK-independent effects, including the amelioration of inflammation, oxidative stress, cellular senescence, apoptotic cell death, and neovascularization [14,15,16]. A previous study showed that the combination of Met and N-acetylcysteine had protective effects against hyperglycemia-induced cardiac deterioration [17]. In addition, Met alone or in combination with the p38 MAPK inhibitor SB203580 provided protection against myocardial ischemia/reperfusion injury in diabetic rodent models and H9c2 cardiomyoblasts [18,19,20,21]. Despite its cardioprotective potential, Met acts as a non-competitive inhibitor of mitochondrial respiratory chain complex I, which decreases the capacity for mitochondrial oxidative phosphorylation [22]. Consequently, under hypoxia or circulatory insufficiency scenarios, Met can increase the risk of lactic acidosis, a condition that mimics or exacerbates the heart-failure-like phenotype.

Recently, imeglimin (Ime), a novel oral antidiabetic agent that shares mechanistic similarities with Met, has emerged in clinical practice. Ime is expected to address several unmet medical needs of patients with T2D using Met or other antihyperglycemic therapies [23,24,25,26,27]. Unlike Met, Ime exerts only a mild inhibitory effect on mitochondrial respiratory chain complex I, which may contribute to a lower risk of lactic acidosis [28]. Interestingly, recent studies have demonstrated that Ime also has beneficial effects on cardiorenal and metabolic dysfunction by inhibiting ROS production [29,30]. However, in contrast to extensively studied Met, the effects of Ime on hyperglycemia-related cellular bioenergetic alterations and oxidative stress in cardiac cells remain unclear. Given that excessive production of ROS is one of the essential mechanisms underlying hyperglycemia-induced cardiac injury, Ime may theoretically exert cardioprotective effects by alleviating ROS overproduction.

In addition to metabolic remodeling and oxidative stress, chronic hyperglycemia induces cellular adhesion disturbances [31,32]. Electrocardiographic abnormalities are frequently observed in patients with T2D [33]. Although diabetes-induced electrophysiological alterations are attributed to the dysregulation of ion channels [34,35,36], the effects of high-glucose-induced stress on intercellular adhesion in cardiomyocytes that form functional syncytia through intercalated discs remain incompletely elucidated. Excessive ROS production under high-glucose conditions may compromise intercellular junctions, potentially establishing a vicious cycle involving metabolic, oxidative, and adhesive dysregulation. These changes can be functionally evaluated using the transepithelial electrical resistance (TEER), an index of intercellular barrier integrity. Given the association among oxidative stress, mitochondrial dysfunction, and cell-to-cell junction stability, antidiabetic agents that attenuate ROS production and modulate mitochondrial respiration may affect cell adhesion. However, it remains unclear whether Met or Ime affects intercellular adhesion in cardiac cells under high-glucose conditions.

Therefore, to elucidate how Ime functions under high-glucose-induced stress responses, its effects were compared with those of Met in H9c2 cardiomyoblasts. We utilized TEER to assess T2D-related cardiac dysfunction markers, cellular metabolic function, ROS generation, autophagy, as a process regulated by oxidative stress, and intercellular binding properties.

## 2. Results

The aim of the current study was to characterize the effects of the newly developed antidiabetic agent Ime on high-glucose-stimulated cellular functional alterations in cardiac cells in comparison with Met, of which the pharmacological effects on the heart have been well established [37,38,39]. Our preceding studies showed that 2 mM concentrations of Met and Ime were optimum concentrations to evaluate their effects on ROS levels, cellular metabolic responses, and cell proliferation in ARPE19 cells [40], melanoma cells [41], and mouse Schwann cells [42] under N-Glu and H-Glu conditions. Similarly, regarding the viability of H9c2 cells exposed to these agents, neither the H-Glu condition alone nor the addition of 2 mM Met or Ime under H-Glu conditions caused significant cytotoxicity compared with N-Glu conditions (Figure 1), and therefore, these concentrations were used throughout the study.

Initially, the pharmacological effects of Met and Ime on essential cellular metabolism, including mitochondrial function and glycolysis, were studied using an extracellular flux analyzer. As shown in Figure 2, mitochondrial functions, such as basal respiration, ATP-linked respiration, and maximal respiration, in H9c2 cells cultured under H-Glu conditions, were significantly elevated compared to those treated under N-Glu conditions (Figure 2A–F). These H-Glu-induced enhancements in mitochondrial functions were significantly reduced by treatment with 2 mM Met or 2 mM Ime (Figure 2A–F), with Ime additionally increasing proton leak, suggesting a potential uncoupling effect (Figure 2D). While Met reduced both the glycolytic capacity and non-glycolytic acidification, Ime increased basal ECAR in H9c2 cells (Figure 2G–J). In addition, both Met and Ime decreased the baseline OCR/ECAR ratio, suggesting a potential shift from oxidative to glycolytic metabolism; however, these changes were not statistically significant under H-Glu conditions (Figure 2K). Based on these collective results, it was speculated that H-Glu exposure modulates mitochondrial activity in H9c2 cells and that treatment with 2 mM Met or Ime may suppress the H-Glu-induced hyperactivation of mitochondrial respiratory function.

To test this hypothesis, the major oxidative-stress-related biological indices, ROS levels, and autophagy were assessed in H9c2 cells under various conditions. As shown in Figure 3, (1) levels of both total cellular ROS (t-ROS) and mitochondrial ROS (m-ROS) tended to be increased under H-Glu conditions compared with under N-Glu conditions, (2) the elevated levels of t-ROS were substantially decreased by 2 mM Ime but not 2 mM Met, and (3) the levels of m-ROS under H-Glu conditions were significantly decreased by 2 mM Ime but not 2 mM Met, suggesting that Ime possessed more potent effects against H-Glu-induced oxidative stress in H9c2 cells than Met (Figure 3A,B). As for the levels of autophagy in H9c2 cells estimated based on the ratios of LC3-I and LC3-II, the LC3-II/LC3-I ratio under H-Glu conditions was significantly lower than that under N-Glu conditions and was not altered by treatment with either 2 mM Met or 2 mM Ime (Figure 4). These results suggested that the inhibitory effect of Ime on ROS overproduction under H-Glu conditions was independent of autophagic activity.

Finally, the additional pharmacological effects of these antidiabetic agents on intercellular binding properties, which may be related to cell-to-cell electrical conduction, were evaluated using TEER measurements. As shown in Figure 5, (1) the TEER value was significantly increased under the H-Glu condition compared to that under the N-Glu condition and (2) the elevated TEER value under the H-Glu condition was markedly enhanced by the presence of 2 mM Ime, but not 2 mM Met (Figure 5). Supporting this finding, the mRNA expression of intercellular junction molecules was altered by the presence of Met and Ime (Figure 6A–C). Furthermore, (1) under a H-Glu condition compared to a N-Glu condition, the mRNA expression of connexin43 and N-cadherin was significantly down-regulated and upregulated, respectively, and that of β-catenin was not altered; (2) the mRNA expression of β-catenin was markedly downregulated by Ime under a H-Glu condition; and (3) H-Glu-induced effects on the mRNA expression of connexin43 and N-cadherin were relatively decreased and enhanced by Met or Ime, respectively. These findings suggest that Ime modulates the intercellular adhesion properties of H9c2 cells under H-Glu conditions more effectively and synergistically than Met, in addition to its more potent antioxidative effects.

## 3. Discussion

In the present study, to explore the pharmacological effects of Ime on high-glucose-induced cellular stress in cardiomyocytes, the metabolic functions, levels of ROS, autophagy, and intercellular binding properties of H9c2 cells were compared between Ime and Met under a H-Glu condition and the following results were observed: (1) significantly elevated mitochondrial respiratory functions under H-Glu conditions were suppressed by both Met and Ime; (2) the increased trends of t-ROS and m-ROS under H-Glu conditions were significantly reduced by Ime but not Met; (3) the H-Glu-induced decrease in autophagic activities assessed by the LC3-II/LC3-I ratio was not significantly affected by treatment with either Met or Ime; (4) the intercellular binding property, evaluated based on TEER, was significantly increased by Ime but not Met. Collectively, these findings suggest that Ime exerts more potent and beneficial effects than Met on H9c2 cells under H-Glu-stimulated stress by decreasing ROS production and fine-tuning excessive mitochondrial respiratory functions, independent of autophagic regulation. In addition, Ime modulated intercellular adhesion, which may contribute to enhanced electrical conduction and cell-to-cell interactions in cardiomyocytes.

Several clinical observational studies have suggested the possible beneficial effect of Met on the failing heart in patients with T2D. In a meta-analysis involving 35,950 T2D patients with heart failure, total mortality in those receiving Met was approximately 22% less than in those not receiving Met [HR: 0.78, (95%: 0.71–0.87)] [43]. Another meta-analysis involving 34,504 patients with T2D and heart failure also showed significant reductions in mortality [HR: 0.80, (95% CI: 0.74–0.87)] and in hospitalization [HR, 0.93: (95% CI: 0.89–0.98)] for patients receiving Met compared with patients receiving sulfonylurea and other antidiabetic agents [44]. In addition to these clinical studies, in vitro studies suggested that Met has beneficial effects on diabetes-induced cardiomyocyte injury. For instance, a previous study using primary human and rat cardiomyocytes showed that Met effectively reduced apoptosis, ROS production, and inflammatory response in a protein phosphatase 2A-dependent manner [45]. Another study showed that the Met-induced activation of AMPK improved autophagy by inhibiting the mTOR pathway, thereby alleviating pyroptosis, an inflammatory form of programmed cell death characterized by caspase activation and the release of proinflammatory mediators, in models of diabetic cardiomyopathy [46]. In contrast to Met, for which the beneficial effects on the myocardium under diabetic conditions have been supported by various studies, including randomized controlled trials, no clinical trials have investigated the effects of Ime on the failing heart in patients with T2D. However, several preclinical and experimental studies using rodent diabetes models have suggested the possible therapeutic effect of Ime on the failing heart [29,47]. Collectively, with the present results focusing on cellular biological functions, Ime might have greater therapeutic potential than Met on the failing heart in patients with T2D. Our findings also highlight the need for clinical trials to evaluate the effect of Ime on various cardiac diseases, including heart failure and arrhythmias, in patients with T2D.

Several lines of evidence suggest that a chronic-hyperglycemia-induced increase in both ROS production and oxidative stress plays an essential role in the development and progression of cardiac function [8,9,10,11]. Met has been reported to protect the myocardium from augmented oxidative stress under various experimental conditions [48,49,50,51]. The cardioprotective effects have been attributed to AMPK-dependent mitochondrial quality control, which protects cardiomyocytes from high-glucose-induced oxidative stress [52]. Another study showed that Met exerts direct cytoprotective effects against hypoxia/reoxygenation injury in cardiac cells via AMPK activation and the concomitant inhibition of c-Jun N-terminal kinase [53]. However, the present findings demonstrated that Met suppressed high-glucose-induced excessive mitochondrial respiratory function but had only a limited effect on ROS production, whereas Ime significantly attenuated both metabolic alterations and total ROS levels, indicating a more potent antioxidative effect under high-glucose conditions. Indeed, several experimental studies have suggested the mechanisms underlying the effects of Ime. For example, a recent study demonstrated that Ime suppressed the high-glucose-induced production of the inflammatory cytokine interleukin-1β by reducing t-ROS levels, resulting in the amelioration of mitochondrial dysfunction in microglia [54]. In addition, another study using a rat model mimicking human metabolic syndrome showed that Ime ameliorated metabolic-syndrome-related cardiac dysfunction by reducing oxidative stress and increasing nitric oxidic bioavailability, independent of glycemic management [29]. Interestingly, our findings showed that Ime increased proton leakage (Figure 2D), suggesting that Ime suppressed mitochondrial ROS overproduction by mitigating the high-glucose-induced increase in mitochondrial membrane potential. This finding may represent one of the mechanisms by which Ime fine-tuned the mitochondrial membrane potential, as previously reported for rat primary hepatocytes [55]. Collectively, these observations support our present findings that Ime, but not Met, alleviates high-glucose-induced ROS overproduction. However, the present study focused primarily on functional outcomes and did not address the underlying mechanisms.

With respect to cell-to-cell adhesion, it has been demonstrated, in an in vitro blood–brain barrier (BBB) model, that high-glucose conditions decrease TEER values and the expression of tight junction proteins [56,57,58]. In contrast to the BBB, a previous study showed that H-Glu conditions promote proliferation without affecting retinal pigment epithelium (RPE) integrity [59]. In line with this phenotype, our recent monolayer model of a putative outer blood–retinal barrier (oBRB) using RPE cells showed that TEER values were substantially elevated under H-Glu conditions compared to those under N-Glu conditions and were further enhanced by Met or Ime, in which the effects of Ime were more potent [40]. Furthermore, the cell densities under the H-Glu condition were much higher than those under the N-Glu condition, and those were further enhanced by Met and Ime, despite the fact that the Met-induced effects were less than those of Ime, suggesting that antidiabetic agents, especially Ime, had potent reinforcement effects on the barrier functions of the RPE by stimulating cell proliferation [40]. Therefore, collective findings suggest that the effects of high-glucose conditions vary exclusively among different tissues [40]. In the current study, the intercellular binding properties of H9c2 cells were relatively and substantially modulated under high-glucose and Ime conditions, respectively. Although the molecular mechanisms by which high glucose and Ime enhance cell adhesion remain unclear, our findings indicate that these effects are accompanied by transcriptional changes in genes related to intercellular adhesions and junctions. Given that the myocardium is terminally differentiated and does not undergo cell division, further in vivo studies using intact heart tissues are warranted to determine whether Ime actually affects cell-to-cell adhesion and electrical conductance in diabetes-induced cardiomyopathy, although it has been reported to not prolong the QRS interval in healthy hearts [60].

In conclusion, the present study demonstrated that treatment with Ime effectively alleviates high-glucose-induced metabolic alterations and ROS overproduction in H9c2 cardiomyoblasts, while enhancing intercellular adhesion, in comparison to Met. These findings may lead to improved electrical conduction coupling in impaired cardiomyocytes. Further in vivo studies using diabetic heart models are warranted to validate the underlying signaling pathways that may contribute to the cardioprotective effects of Ime against diabetes-induced cardiac dysfunction.

## 4. Materials and Methods

### 4.1. Planar Culture of H9c2 Cells

Planar cultures of H9c2 rat cardiomyoblasts, purchased from the ATCC (American Type Culture Collection), were processed in two-dimensional (2D) culture dishes (150 mm diameter) using Dulbecco’s Modified Eagle Medium (DMEM) (Wako, Osaka, Japan) supplemented with 10% fetal bovine serum (Biosera, Cholet, France), 1% L-glutamine (Wako, Osaka, Japan), and 1% antibiotic-antimycotic (Thermo Fisher Scientific, Tokyo, Japan) under standard conditions (37 °C, 5% CO_2_), as shown in a previous study [61]. To evaluate the pharmacological effects of Met or Ime on the biological functions of H9c2 cells under different glucose conditions, high glucose (50 mM, H-Glu) and normal physiological glucose (5.5 mM, N-Glu), 2 mM of Ime or Met was incubated with H9c2 cells under H-Glu or N-Glu conditions for 24 hrs until the measurements described below.

### 4.2. Cell Viability Assay

Cell viability was determined using a commercially available kit (Cell Counting Kit-8, CCK-8; DOJINDO, Kumamoto, Japan). In brief, H9c2 cells were seeded at 5000 cells per well in an ordinary 96-well culture plate under standard conditions (37 °C, 5% CO_2_). The cells were then cultured for 24 h in DMEM containing either N-Glu (5.5 mM) or H-Glu (50 mM) supplemented with either DMSO as a vehicle control, 2 mM Met, or 2 mM Ime under the same standard conditions. On the day of the assay, following an incubation of cells with the reactive solutions for 2 h, the absorbance at 450 nm was detected using a microplate reader (multimode plate reader EnSpire^®^, PerkinElmer, MA, USA).

### 4.3. Seahorse Extracellular Flux Analysis

A total of 20,000 H9c2 cells were plated onto Seahorse XFe96 Cell Culture Microplates (Agilent Technologies, #103794-100, Santa Clara, CA, USA) and cultured under N-Glu (5.5 mM) or H-Glu (50 mM) conditions with 2 mM Met, 2 mM Ime, or vehicle control containing dimethyl sulfoxide (DMSO) for 24 hrs. On the day of measurement, the culture medium was exchanged with 180 μL of the same Seahorse XF DMEM assay medium (Agilent Technologies, #103575-100, Santa Clara, CA, USA) supplemented with 5.5 mM glucose, 2.0 mM glutamine, and 1.0 mM sodium pyruvate for all groups to evaluate the effects of prior high-glucose-induced stress under standardized conditions. The plates were then placed in a CO_2_-free incubator at 37 °C for 1 hr prior to the measurements. The oxygen consumption rate (OCR) and extracellular acidification rate (ECAR) were simultaneously assessed using a Seahorse XFe96 Bioanalyzer (Agilent Technologies) according to the manufacturer’s guidelines. OCR and ECAR were measured at baseline and following the sequential injection of oligomycin (2.0 μM), carbonyl cyanide p-trifluoromethoxyphenylhydrazone (FCCP, 5.0 μM), a mixture of rotenone and antimycin A (1.0 μM), and 2-deoxyglucose (2DG, 10 mM). 2DG was injected to confirm that the residual OCR was non-mitochondrial and not driven by the glycolysis-dependent oxygen demand. Each measurement cycle consisted of 3 min of mixing followed by 3 min of signal acquisition. Upon completion of the run, total protein content per well was determined using a BCA protein assay (TaKaRa Bio, Shiga, Japan) for data normalization.

Several mitochondrial and glycolytic parameters, including the basal respiration, ATP-linked respiration, proton leak, maximal respiration, non-mitochondrial respiration, basal ECAR, glycolytic reserve, non-glycolytic acidification, and baseline OCR/ECAR ratio, were calculated according to previously described methods [62].

### 4.4. Measurement of Levels of Reactive Oxygen Species (ROS)

Levels of total cellular ROS (t-ROS) and mitochondrial ROS (m-ROS) in H9c2 cells were determined using commercially available kits (t-ROS: ROS Assay Kit-Highly Sensitive DCFH-DA, m-ROS: MitoBright ROS Deep Red-mitochondrial Superoxide Detection, DOJINDO, Kumamoto, Japan) according to the manufacturer’s instructions. In brief, a total of 20,000 H9c2 cells were seeded in a 96-well clear-bottom black plate (96 Well Black/Clear Bottom Plate, TC Surface, Thermo Scientific™, Waltham, MA, USA) and cultured in DMEM containing either N-Glu (5.5 mM), H-Glu (50 mM), H-Glu with 2 mM Met, or H-Glu with 2 mM Ime for 24 hrs under standard conditions (37 °C, 5% CO_2_). After treatment, the medium was removed, cells were washed twice with Hanks’ Balanced Salt Solution (HBSS) and 100 μL of Working Solution for t-ROS, or m-ROS was added to the cells and incubated for 30 min at 37 °C in 5% CO_2_. The Working Solution was then discarded, cells were washed twice with HBSS, and 100 μL of fresh HBSS was added for the measurement. t-ROS signals were normalized to the total protein per well measured based on a BCA protein assay (TaKaRa Bio, Shiga, Japan). m-ROS signals were normalized to the mitochondrial content by dividing the Deep Red fluorescence by the corresponding fluorescence of MitoTracker Green FM (Cell Signaling Technology, #9074S, Danvers, MA, USA) for each well. Fluorescence intensities were measured on a microplate reader using the following channels: t-ROS (DCFH-DA): excitation: 485 nm, emission: 535 nm, m-ROS (MitoBright ROS Deep Red): excitation: 635 nm, emission: 710 nm, mitochondrial content (MitoTracker Green FM): excitation 488 nm and emission 515 nm.

### 4.5. Measurement of TEER

TEER values for the monolayers of H9c2 cells were measured by using a commercially available TEER plate (pore size of 0.4 μm, diameter of 12 mm; Corning Transwell 12-well plate, Sigma-Aldrich, St. Louis, MO, USA) with an electrical resistance system (KANTO CHEMICAL CO. INC., Tokyo, Japan) as previously reported [63,64]. In brief, H9c2 cells were seeded at a density of 10,000 cells per well and cultured under standard conditions (37 °C, 5% CO_2_). After five days, the culture medium was replaced with fresh DMEM containing one of the following conditions: N-Glu (5.5 mM), H-Glu (50 mM), H-Glu with 2 mM Met, or H-Glu with 2 mM Ime. TEER measurements were performed after 24 h of treatment for each condition.

### 4.6. Western Blotting

Cells were lysed on ice using CelLytic™ MT Cell Lysis Reagent (#C3228, Sigma-Aldrich, St. Louis, MO, USA) supplemented with a protease inhibitor cocktail (#786-437, G-Biosciences, St. Louis, MO, USA). The lysates were centrifuged at 12,000× *g* for 15 min at 4 °C, and the resulting supernatants were collected. Protein concentrations of the lysates were determined using the BCA protein assay kit (TaKaRa Bio, Shiga, Japan). Equal amounts of protein were separated on 12.5% polyacrylamide gels and transferred onto PVDF membranes (Millipore, Bedford, MA, USA.). Membranes were blocked with Tris-buffered saline containing 5% skim milk and 0.05% Tween-20 (TBS-T) for 1 h at room temperature followed by overnight incubation at 4 °C with primary antibodies: anti-LC3 (1:1000, #4108, Cell Signaling Technology, Beverly, MA, USA) and anti-α-tubulin (1:1000, #2144, Cell Signaling Technology, Beverly, MA, USA). After washing with TBS-T, membranes were incubated for 1 h at room temperature with HRP-conjugated anti-rabbit secondary antibody (1:5000, NA934V, Cytiva, Marlborough, MA, USA), and bands were visualized using the Immobilon Western Detection Kit (Millipore, Billerica, MA, USA). Membranes were imaged and processed digitally.

### 4.7. Quantitative Real-Time PCR

The extraction of total RNA, reverse transcription, and quantitative real-time PCR (qRT-PCR) were performed as described in our previous reports [65,66], using specific primers and probes (Appendix A).

### 4.8. Statistics

All statistical analyses were carried out using GraphPad Prism 8 or 9 (GraphPad Software, San Diego, CA, USA). Data are presented as the mean ± standard error of the mean (SEM), with individual data points displayed in the plots where applicable. Comparisons among multiple groups were assessed using one-way analysis of variance (ANOVA) followed by Tukey’s Honestly Significant Difference (HSD) post-hoc test. A two-tailed *p*-value < 0.05 was considered statistically significant.

## Figures and Tables

**Figure 1 ijms-26-08913-f001:**
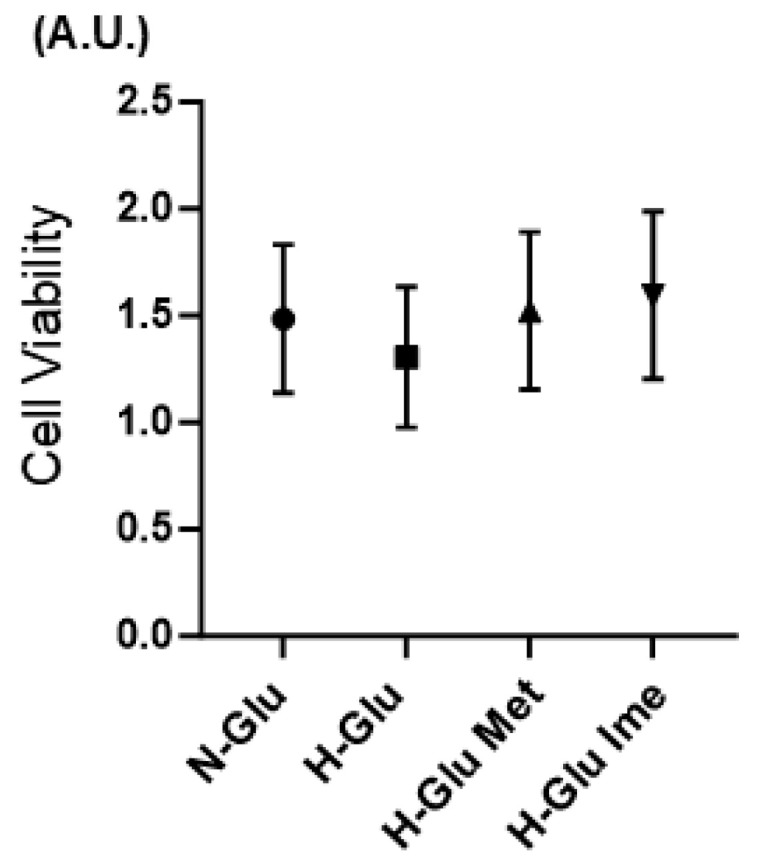
Met- or Ime-induced cytotoxicity in planner cultured H9c2 cells under high-glucose conditions. The cytotoxicity of 2 mM metformin (Met) and 2 mM imeglimin (Ime) towards H9c2 cells under high-glucose (50 mM, H-Glu) conditions was assessed and compared with that under normal glucose (5.5 mM, N-Glu) conditions. Viable cells in planar cultured H9c2 cells were evaluated using a commercially available kit (Cell Counting Kit-8, Dojin Laboratories, Kumamoto, Japan), and the values were plotted (*n* = 3). Data are expressed in arbitrary units (A.U.).

**Figure 2 ijms-26-08913-f002:**
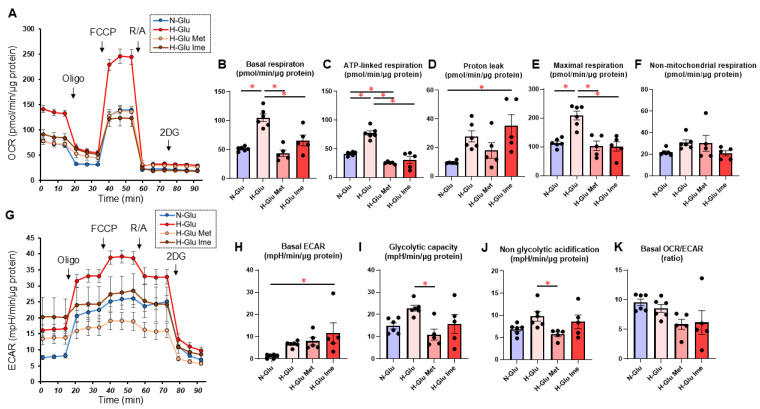
Effects of Met or Ime on the cellular metabolic functions of H9c2 cells under high-glucose conditions. H9c2 cells were cultured under normal-glucose (5.5 mM, N-Glu) conditions and then incubated for 24 hrs under N-Glu or high-glucose (50 mM, H-Glu) conditions, with or without 2 mM metformin (Met) or 2 mM imeglimin (Ime). Metabolic function was assessed using a Seahorse XFe96 Bioanalyzer. Panel (**A**): Representative oxygen consumption rate (OCR) traces. Panel (**B**): Basal respiration, calculated as OCR at baseline minus OCR after R/A injection. Panel (**C**): ATP-linked respiration, calculated as OCR at baseline minus OCR after Oligo injection. Panel (**D**): Proton leak, calculated as OCR after Oligo injection minus OCR after R/A injection. Panel (**E**): Maximal respiration, calculated as OCR after FCCP injection minus OCR after R/A injection. Panel (**F**): Non-mitochondrial respiration, represented as OCR after R/A injection. Panel (**G**): Representative extracellular acidification rate (ECAR) traces. Panel (**H**): Basal ECAR, calculated as ECAR at baseline minus ECAR measured at the final data point following 2DG injection. Panel (**I**): Glycolytic capacity, calculated as ECAR after Oligo injection minus ECAR measured at the final data point following 2DG injection. Panel (**J**): Non-glycolytic acidification, represented as ECAR measured at the final data point following 2DG injection. Panel (**K**): Baseline OCR/ECAR ratio (OCR at baseline)/(ECAR at baseline). Oligo: oligomycin; FCCP: carbonyl cyanide p-trifluoromethoxyphenylhydrazone; R/A: rotenone/antimycin A; 2DG: 2-deoxyglucose. All experiments were performed using fresh preparations (*n* = 5–6). * *p* < 0.05.

**Figure 3 ijms-26-08913-f003:**
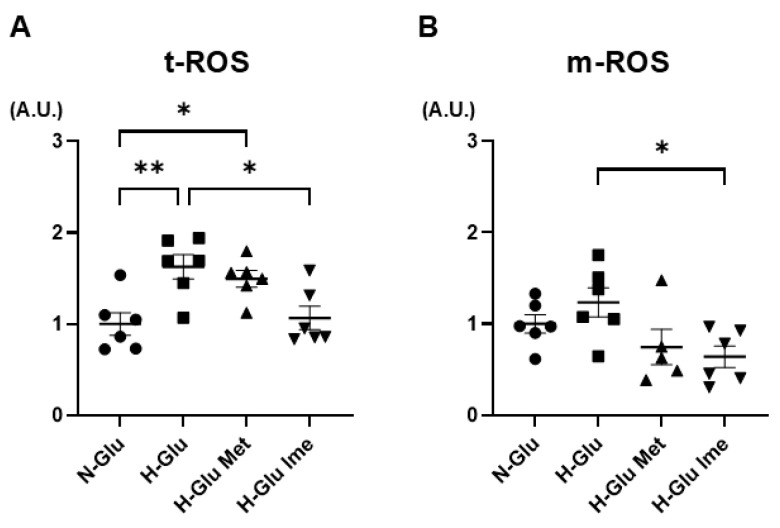
H9c2 cells were cultured under normal-glucose (5.5 mM, N-Glu) conditions and then incubated for 24 hrs under N-Glu or high-glucose (50 mM, H-Glu) conditions, with or without 2 mM metformin (Met) or 2 mM imeglimin (Ime). Reactive oxygen species (ROS) production was assessed using DCFH-DA. Panel (**A**): Total ROS production (t-ROS). Panel (**B**): mitochondrial ROS production (m-ROS). All experiments were performed using fresh preparations (*n* = 7). Values are expressed in arbitrary units (A.U.). * *p* < 0.05, ** *p* < 0.01.

**Figure 4 ijms-26-08913-f004:**
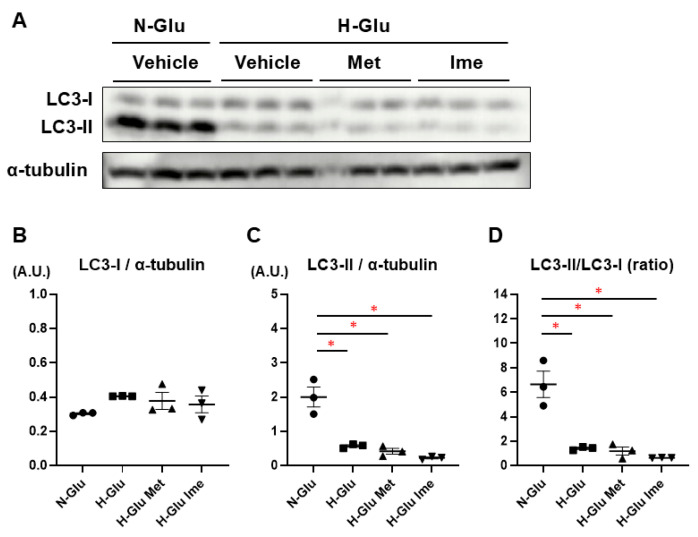
Evaluation of autophagy-related LC3-I and LC3-II expression levels via Western blotting. Lysates from H9c2 cells cultured under normal-glucose (N-Glu), high-glucose (H-Glu), high-glucose-with-metformin (Met), and high-glucose-with-imeglimin (Ime) conditions were subjected to Western blot analysis. Panel (**A**): Representative Western blot images showing the expression levels of LC3-I, LC3-II, and the loading control α-tubulin in H9c2 cells cultured under normal-glucose (N-Glu) or high-glucose (H-Glu) conditions with or without metformin (Met, 2 mM) or imeglimin (Ime, 2 mM) for 24 hrs. Panel (**B**): Quantification of LC3-I expression. Panel (**C**): Quantification of LC3-II expression. Expression levels of the LC3-I. Panel (**D**): Quantification of LC3-II/LC3-I ratio. Data are expressed as relative values in arbitrary units (A.U.) or as ratios. * *p* < 0.05.

**Figure 5 ijms-26-08913-f005:**
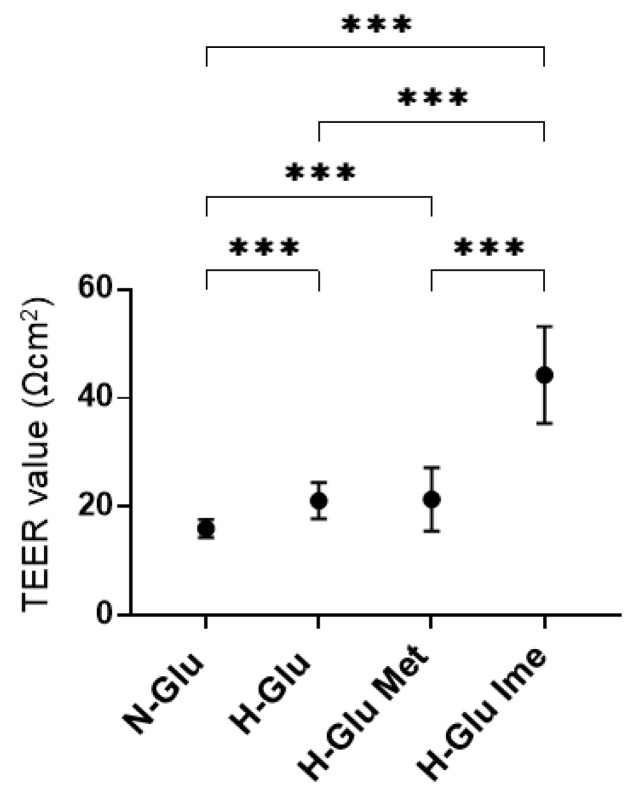
Effects of Met or Ime on transepithelial electrical resistance (TEER) values of planar H9c2 cell monolayers. H9c2 cells were cultured under normal-glucose (5.5 mM, N-Glu) conditions and then incubated for 24 h under N-Glu or high-glucose (50 mM, H-Glu) conditions, with or without 2 mM metformin (Met) or 2 mM imeglimin (Ime). H9c2 cell monolayers were then subjected to electric resistance (Ωcm^2^) measurements using transepithelial electrical resistance (TEER), and the values were plotted. All experiments were performed in triplicate using fresh preparations (*n* = 3). *** *p* < 0.005.

**Figure 6 ijms-26-08913-f006:**
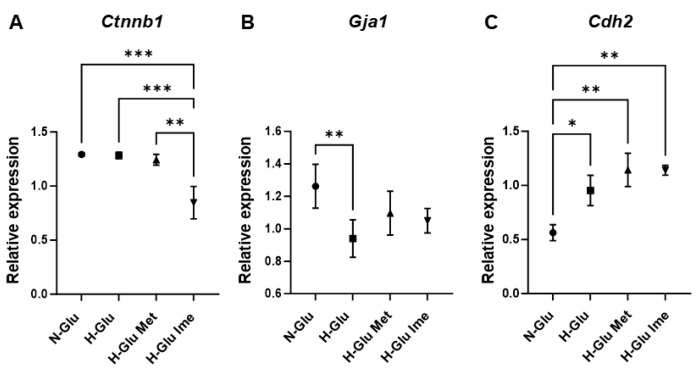
H9c2 cells were cultured under normal-glucose (5.5 mM, N-Glu) conditions and then incubated for 24 h under N-Glu or high-glucose (50 mM, H-Glu) conditions, with or without 2 mM metformin (Met) or 2 mM imeglimin (Ime). Each sample was subjected to qRT-PCR analysis, and the mRNA expression levels of β-catenin (Ctnnb) Panel (**A**), connexin 43 (Gja1), Panel (**B**) and N-cadherin (Cdh2) Panel (**C**) were estimated. All experiments were performed in duplicate using fresh preparations (*n* = 5 each). * *p* < 0.05, ** *p* < 0.01, *** *p* <0.005.

## Data Availability

The original contributions presented in this study are included in the article/Appendix A; further inquiries can be directed to the corresponding author.

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
