# Peer review of "Imeglimin Alleviates High-Glucose-Induced Bioenergetic and Oxidative Stress Thereby Enhancing Intercellular Adhesion in H9c2 Cardiomyoblasts"

_ijms, 2025, doi:10.3390/ijms26188913_

Round 1

Reviewer 1 Report

Comments and Suggestions for Authors

The authors have presented an interesting manuscript where they have compared the effect of imeglimin (Ime, 2 mM) and metformin (Met, 2 mM) in H9c2 cardiomyoblasts exposed to high glucose. The authors have introduced the topic and have put a clear hypothesis for the paper. But the following concerns need to be addressed before accepting the manuscript. These comments are meant to make the manuscript better. 

General comments
Please soften “enhances intercellular adhesion” unless supported by stronger protein and functional data.

What is the effect of the Metformin and IM treatment on the cells under normoxic conditions? 

Figure 2, in the methods section, the authors have mentioned that the assay buffer contained 5.5 mM glucose for all groups, indicating all the measurements were effectively normoglycemic. How was the high glucose introduced into the culture system? Were the cells pretreated with high glucose, and then the measurement was conducted in low glucose concentration? If that's the case, these experiments need to be repeated with matched glucose (5.5 vs 50 mM) and drugs injected using the first port in the sehorse. 
Is 2DG in the OCAR measurements a typo, or was it introduced in the system for a particular reason?

Authors should show individual data points in Figure 3. 

The findings in Figure 3 conflict with earlier reports of metformin inactivating Complex I, since Complex I activity is a key source of mitochondrial ROS, and metformin is known to inhibit Complex I. Therefore, a decrease in mitochondrial ROS and overall cellular ROS levels in cells treated with metformin should have been observed. The authors need to provide a detailed discussion of these unexpected results. 

Since dye-based quantification with probes such as MitoBright ROS Deep Red can be influenced by differences in overall mitochondrial content, it would be helpful to normalize the data to a mitochondrial marker. This could be done using either a genetically encoded fluorescent mitochondrial marker or a dye independent of potential like MitoTracker Green. Including such normalization would strengthen the conclusions and provide clearer insight into mitochondrial ROS levels. Adding a positive control, like rotonone, and a negative control, like mitoTEMPO, will significantly enhance the interpretation of data. 

Authors have misclassified N-cadherin, which is an adherens-junction protein, not a tight-junction protein. Connexin43 is a gap junction; β-catenin is an adherens-junction scaffold. The convention in the field is to check the protein levels of these marker proteins. β-catenin mRNA is downregulated by Ime, which should weaken adhesion, yet the authors have claimed the opposite.

Author Response

Dear Editor and Reviewers,

We are very grateful for your supportive and constructive comments on our manuscript entitled “Imeglimin alleviates high glucose-induced bioenergetic and oxidative stress thereby enhancing intercellular adhesion in H9c2 cardiomyoblasts”. We have carefully addressed all the reviewers’ comments and have prepared a revised version of our manuscript accordingly. The changes are summarized below.

Editorial comment:

Thank you very much for handling our paper carefully. According to the editorial comment, we have reduced the similarities and have additionally corrected several typographical and formatting errors as highlighted.

Reviewer 1 comments:

  1. General comments: Please soften “enhances intercellular adhesion” unless supported by stronger protein and functional data.

Response: Thank you very much for your careful review. We totally agree that “enhance” was overstatement without supporting protein-based experiments. In the revised menauscript, we have accordingly rephrased this phrase with “modulates” or “alter” in Abstract, 4th paragraph of the Results section, and 1st paragraph of the Discussion section of the revised manuscript.

  1. What is the effect of the Metformin and Imeglimin treatment on the cells under normoxic conditions?

Response: We sincerely appreciate the reviewer’s valuable comment and apologize for the confusion caused by our inappropriate use of the term “normoxia.” In this study, we did not manipulate oxygen levels, and all experiments were performed under standard culture conditions (37℃, 5% CO2, and 21% Oâ‚‚). To avoid misunderstanding, we have replaced “normoxia” with “standard culture conditions” throughout the manuscript.

  1. Figure 2, in the methods section, the authors have mentioned that the assay buffer contained 5.5 mM glucose for all groups, indicating all the measurements were effectively normoglycemic. How was the high glucose introduced into the culture system? Were the cells pretreated with high glucose, and then the measurement was conducted in low glucose concentration? If that's the case, these experiments need to be repeated with matched glucose (5.5 vs 50 mM) and drugs injected using the first port in the Seahorse. Is 2DG in the OCAR measurements a typo, or was it introduced in the system for a particular reason?

Response: We thank the reviewer for these important comments and apologize for the insufficient explanation regarding our experimental design.

Regarding the first question, H9c2 cells were pre-exposed for 24 hrs to either normal glucose (5.5 mM) or high glucose (50 mM) medium with or without metformin or imeglimin. On the assay day, the culture medium was replaced with the same Seahorse assay buffer containing 5.5 mM glucose for all groups, thereby standardizing measurement conditions. The aim of this design was to evaluate the effects of “prior high-glucose stress” on mitochondrial and glycolytic functions while minimizing the confounding influence of acute substrate differences during the Seahorse run. Indeed, in our preliminary experiments, we previously tested Seahorse assays using 50 mM glucose in the assay buffer. However, we found that both baseline OCR and ECAR values became markedly elevated and saturated, leading to reduced dynamic range and unreliable data interpretation. In the revised manuscript, we have revised the Methods to clarify these points.

Regarding the second question, the inclusion of 2-deoxy-D-glucose (2DG) in the OCR measurements was intentional and not a typo. 2DG was injected after Rotenone and Antimycin A to confirm that the residual OCR observed after mitochondrial complex inhibition was non-mitochondrial and not driven by glycolysis-dependent oxygen demand. As expected, 2DG caused no additional decrease in OCR, confirming that the remaining oxygen consumption represents non-mitochondrial processes. We have clarified this purpose in the revised Methods section, and we have also amended revised Figure 2 to accurately reflect the position of the 2DG injection.

  1. Authors should show individual data points in Figure 3.

Response: We sincerely appreciate your valuable comment. We totally agree this suggestion and we have now added individual data points to revised Figure 3. Furthermore, we would like to note that Figure 3 has been updated because we performed additional experiments to address the reviewer’s subsequent question (Comment Reviewer #1-5). While the overall conclusions remain unchanged, some data values in Figure 3 have been revised to reflect the newly obtained results.

  1. The findings in Figure 3 conflict with earlier reports of metformin inactivating Complex I, since Complex I activity is a key source of mitochondrial ROS, and metformin is known to inhibit Complex I. Therefore, a decrease in mitochondrial ROS and overall cellular ROS levels in cells treated with metformin should have been observed. The authors need to provide a detailed discussion of these unexpected results. Since dye-based quantification with probes such as MitoBright ROS Deep Red can be influenced by differences in overall mitochondrial content, it would be helpful to normalize the data to a mitochondrial marker. This could be done using either a genetically encoded fluorescent mitochondrial marker or a dye independent of potential like MitoTracker Green. Including such normalization would strengthen the conclusions and provide clearer insight into mitochondrial ROS levels. Adding a positive control, like rotenone, and a negative control, like mitoTEMPO, will significantly enhance the interpretation of data.

Response: We sincerely appreciate this excellent comment. We totally agree with your suggestion. From the perspective of proper normalization, we repeated the following experiments and reanalyzed the data accordingly.

1) Total cellular ROS (t-ROS) levels were normalized to “protein contents”, and the results remained unchanged.

2) For mitochondrial ROS (m-ROS), we performed double staining with MitoTracker Green to normalize the data to “mitochondrial content”. After normalization, the Met-associated increase in m-ROS levels disappeared, whereas the Ime-induced decrease in m-ROS levels remained significant. Although we were not able to perform experiments using positive and negative controls, we used validated reagents according to the manufacturer’s instructions, where these controls were tested and confirmed (https://www.dojindo.com/manual/MT16/). The manuscript as well as Figure 3 has been updated to reflect these findings.

  1. Authors have misclassified N-cadherin, which is an adherens-junction protein, not a tight-junction protein. Connexin43 is a gap junction; β-catenin is an adherens-junction scaffold. The convention in the field is to check the protein levels of these marker proteins. β-catenin mRNA is downregulated by Ime, which should weaken adhesion, yet the authors have claimed the opposite.

Response: We sincerely appreciate your excellent comment and suggestion. As the reviewer currently pointed out, these three genes we analyzed represent only a subset of molecules involved in intercellular adhesion. We agree that it was an overstatement to conclude “increased cell adhesion” based solely on the analysis of these three genes. We did not evaluate the specific functions related to tight junctions, gap junctions, or adhesion scaffold. Therefore, we have toned down the description regarding the Ime-associated alterations in intercellular adhesion throughout the revised manuscript.

Reviewer 2 comments

  1. When describing the statistical analysis (lines 384-386), the authors do not provide detailed information, but refer to previous works (https://doi.org/10.1038/s41598-020-64674-1, https://doi.org/10.1167/iovs.61.6.13). However, in these works, the descriptions of statistical methods differ and it is not entirely clear what was used in this study. It is better to describe the data processing methodology in detail in the text.

Response: We sincerely appreciate your excellent comment. We totally agree that we should represent detailed information related to the statistical analysis. In the revised manuscript, we have created a new “Statistics” section in the Materials and Methods section specifically describing the data processing and statistical analysis methodology used in this study.

  1. In the description of the cell culturing method (lines 310-319), the manufacturers of reagents are not indicated.

Response: Thank you very much for your careful review. As pointed out, corresponding information has been included in the revised manuscript.

  1. In the description of the method for assessing cell viability (lines 320-324), the number of cells seeded for analysis and the type of culture vessel used are not indicated.

Response: We sincerely appreciate your careful review. As suggested, the number of cells seeded and the type of culture vessel for the present experiments have been included in the revised manuscript.

  1. In addition, it would be interesting to know why the authors of the article chose 2 mM concentrations of imeglimin and metformin for the in vitro experiments.

Response: Thank you very much for this excellent comment. In terms of the reason for choosing 2 mM concentrations of Met and Ime, our preceding studies demonstrated that 2 mM concentrations of Met and Ime were optimum concentrations to evaluate their effects on ROS levels, cellular metabolic responses and cell proliferation in ARPE19 cells (doi: 10.3390/bioengineering12030265), melanoma cells (doi: 10.3390/ijms26031014) and mouse Schwann cells (doi: 10.3389/fncel.2025.1634262). In addition, drug-induced cytotoxic effects of Met and Ime were at negligible levels as shown in Figure 1. This information has been included in the 1st paragraph of the revised Result section to clarify the rationale.

Reviewer 2 Report

Comments and Suggestions for Authors

The central problem of the work is a comparative analysis of the effect of the new antidiabetic drug imeglimin and traditional metformin on cellular stress in hyperglycemia. The study is aimed at assessing their impact on metabolic parameters, oxidative stress and intercellular adhesion. The scientific significance of the study is determined by the growing problem of diabetic cardiomyopathy and the need to find new approaches to the treatment of this disease.

The novelty of the work lies in studying the effects of imeglimin at the cellular level in the culture of cardiomyoblasts. The methodological approach of the authors includes an assessment of the cytotoxicity of the drugs under study, as well as a detailed analysis of changes in the metabolic functions of the cell and intercellular interactions in response to drug exposure in vitro.

The results obtained are fully illustrated by figures and tables, the conclusions of the study generally correspond to the data obtained. The list of references is sufficient, includes 64 sources, about half of which (28) were published in the last 5 years.

Сomments and questions:

When describing the statistical analysis (lines 384-386), the authors do not provide detailed information, but refer to previous works (https://doi.org/10.1038/s41598-020-64674-1, https://doi.org/10.1167/iovs.61.6.13). However, in these works, the descriptions of statistical methods differ and it is not entirely clear what was used in this study. It is better to describe the data processing methodology in detail in the text.

In the description of the cell culturing method (lines 310-319), the manufacturers of reagents are not indicated.

In the description of the method for assessing cell viability (lines 320-324), the number of cells seeded for analysis and the type of culture vessel used are not indicated.

In addition, it would be interesting to know why the authors of the article chose 2mM concentrations of imeglimin and metformin for the in vitro experiments.

Author Response

(The authors gave the same response as above.)
